# Mild-moderate CKD is not associated with cognitive impairment in older adults in the Alzheimer's Disease Neuroimaging Initiative cohort

**Aditi Gupta**[1,2,3]*, **Kevin Kennedy**[4], **Jaime Perales-Puchalt**[3,5], **David Drew**[6], **Srinivasan Beddhu**[7], **Mark Sarnak**[6], **Jeffrey Burns**[3,5], **the Alzheimer's Disease Neuroimaging Initiative**[¶]

**1** The Kidney Institute, University of Kansas Medical Center, Kansas City, KS, United States of America, **2** Division of Nephrology and Hypertension, University of Kansas Medical Center, Kansas City, KS, United States of America, **3** Alzheimer's Disease Center, University of Kansas Medical Center, Kansas City, KS, United States of America, **4** Kansas City Veterans Affairs, University of Kansas Medical Center, Kansas City, KS, United States of America, **5** Department of Neurology, University of Kansas Medical Center, Kansas City, KS, United States of America, **6** Division of Nephrology and Hypertension, Tufts Medical Center, Boston MA, United States of America, **7** University of Utah, Salt Lake City, UT, United States of America

¶ A complete membership of the author group can be found in the Acknowledgments.
* agupta@kumc.edu

**Data Availability Statement:** Data used in preparation of this article were obtained from the Alzheimer's Disease Neuroimaging Initiative (ADNI)

## Abstract

### Background

Chronic kidney disease (CKD) is associated with cognitive impairment and dementia. We examined whether this relationship hold true in older adults, who have a higher prevalence of both CKD and dementia.

### Design, setting, participants, and measurements

We conducted a cross-sectional secondary analysis of an established observational cohort. We analyzed data from the Alzheimer's Disease Neuroimaging Initiative (ADNI), an NIH funded, multicenter longitudinal observational study, which includes participants with normal and impaired cognition and assesses cognition with a comprehensive battery of neuropsychological tests. We included a non-probability sample of all ADNI participants with serum creatinine measurements at baseline (N = 1181). Using multivariable linear regression analysis, we related the CKD Epidemiology Collaboration equation eGFR with validated composite scores for memory (ADNI-mem) and executive function (ADNI-EF).

### Results

For the 1181 ADNI participants, the mean age was 73.7 ± 7.1 years. Mean estimated glomerular filtration rate (eGFR) was 76.4 ± 19.7; 6% had eGFR<45, 22% had eGFR of 45 to <60, 51% had eGFR of 60–90 and 21% had eGFR>90 ml/min/1.73 $m^2$. The mean ADNI-Mem score was 0.241 ± 0.874 and mean ADNI-EF score was 0.160 ± 1.026. In separate multivariable linear regression models, adjusted for age, sex, race education and BMI, there

database (adni.loni.usc.edu). The ADNI data is easily accessible to any investigator after approval by the ADNI committee. Other investigators can reach out to the corresponding author once they have permission from ADNI to access data, and the study team can then assist them.

**Funding:** This work is supported by NIH K23 AG055666 (AG) Cognitive Impairment in End Stage Renal Disease, NIH CTSA grant UL1 TR000001 (KUMC), and NIH P30 AG035982 (KU ADC). Data collection and sharing for this project was funded by the Alzheimer's Disease Neuroimaging Initiative (ADNI) (National Institutes of Health Grant U01 AG024904) and DOD ADNI (Department of Defense award number W81XWH-12-2-0012). ADNI is funded by the National Institute on Aging, the National Institute of Biomedical Imaging and Bioengineering, and through generous contributions from the following: AbbVie, Alzheimer's Association; Alzheimer's Drug Discovery Foundation; Araclon Biotech; BioClinica, Inc.; Biogen; Bristol-Myers Squibb Company; CereSpir, Inc.; Cogstate; Eisai Inc.; Elan Pharmaceuticals, Inc.; Eli Lilly and Company; EuroImmun; F. Hoffmann-La Roche Ltd and its affiliated company Genentech, Inc.; Fujirebio; GE Healthcare; IXICO Ltd.; Janssen Alzheimer Immunotherapy Research & Development, LLC.; Johnson & Johnson Pharmaceutical Research & Development LLC.; Lumosity; Lundbeck; Merck & Co., Inc.; Meso Scale Diagnostics, LLC.; NeuroRx Research; Neurotrack Technologies; Novartis Pharmaceuticals Corporation; Pfizer Inc.; Piramal Imaging; Servier; Takeda Pharmaceutical Company; and Transition Therapeutics. The Canadian Institutes of Health Research is providing funds to support ADNI clinical sites in Canada. Private sector contributions are facilitated by the Foundation for the National Institutes of Health (www.fnih.org). The grantee organization is the Northern California Institute for Research and Education, and the study is coordinated by the Alzheimer's Therapeutic Research Institute at the University of Southern California. ADNI data are disseminated by the Laboratory for Neuro Imaging at the University of Southern California The funders had no role in study design, data collection and analysis, decision to publish, or preparation of the manuscript.]

**Competing interests:** The authors declare no conflict of interest that alters their adherence to PLOS ONE policies on sharing data and materials. AG has a consultancy agreement with Novartis Pharmaceuticals and has grant funding from Novartis and Veloxis Pharmaceuticals; none of which are relevant to the current manuscript.

was no association between each 10 ml/ min/1.73 m$^2$ higher eGFR and ADNI-Mem (β -0.02, 95% CI -0.04, 0.02, p = 0.56) or ADNI-EF (β 0.01, 95% CI -0.03, 0.05, p = 0.69) scores.

## Conclusion

We did not observe an association between eGFR and cognition in the older ADNI participants.

## Introduction

Cognitive impairment and dementia negatively affect activities of daily living, quality of life, sense of well-being, morbidity and mortality [1–4]. Age is an independent risk factor for both cognitive impairment and chronic kidney disease (CKD) and increase in life expectancy has fueled an unprecedented growth in the prevalence of both conditions [5–7]. With the current threshold of glomerular filtration rate (GFR) <60 mL/min/1.73 m$^2$ for CKD approximately one-half of older adults >70 years have CKD [7]. The impact of a lower GFR in older adults is controversial [8–11] as lower GFR in older adults is associated with different renal pathology [12, 13] as well as clinical outcomes [14–18] when compared to younger persons. Age associated decline in GFR is less likely to progress to end stage kidney disease (ESKD), and most older people with CKD die with CKD rather than from it [18].

In older adults, the association of eGFR and cognition remains contentious. While some studies report absence of cognitive impairment at eGFR of >30 mL/min/1.73 m$^2$ [19–22], others report incremental risk of cognitive impairment with declining eGFR [23–25]. Baseline differences in age and comorbidities or use of tests designed for screening and not detection of severity of cognitive impairment make the results difficult to interpret. For example, in The Reasons for Geographic and Racial Differences in Stroke (REGARDS) study [23] the mean age of participants with CKD was 71 years compared to 64 years in controls. With an individual's risk of developing dementia doubling every 5 years after age 65, matching for age at baseline is important. There were also baseline differences in other confounding variables such as sex, education, and comorbidities. Similar baseline differences were present in the Chronic Renal Insufficiency Cohort Cognitive Study (CRIC-COG) [25]. In fact, a prospective cohort study indicated association of dementia and low eGFR in unadjusted analysis, but this association was lost after adjustment for confounding variables, highlighting the importance of these baseline confounders [26]. Another study in older men [27] did not show an association of lower eGFR with cognition after adjusting for differences in age, race and education. In the Health, Aging, and Body Composition (Health ABC) Study [28] after stratification by age, there was no association between eGFR and cognition in the older half of the participants indicating an interaction of age in this association. Some other studies included patients with moderate to severe CKD [24] and not mild CKD, which is more common in older adults.

Thus, it remains unclear if the association between lower GFR and cognitive impairment holds true for the older adults with CKD, where estimation of kidney function may be less accurate and the implications of a lower GFR may be different. With paucity of data on cognitive outcomes with a lower GFR in older adults, risk estimation and guidelines for management, counselling and education on cognitive impairment and dementia for older patients are lacking. Overestimation of risk of cognitive impairment can lead to unnecessary anxiety and

over utilization of health resources while underestimation can lead to less aggressive preventative management and increase in rates of dementia.

To better understand the association between mild-moderate CKD and cognition, we conducted a post hoc analysis of data from the Alzheimer's Disease Neuroimaging Initiative (ADNI). The Alzheimer's Disease Neuroimaging Initiative (ADNI) is an NIH supported multicenter, longitudinal, prospective, observational study of normal cognitive aging, mild cognitive impairment (MCI), and early dementia. We chose the ADNI cohort for this analysis as the ADNI has an older population with a mean age of 74 years without a significant burden of medical issues, since that was an exclusion criterion for the ADNI. Moreover, the ADNI provided comprehensive neuropsychological testing with detailed evaluation of memory and executive function, two domains of cognition preferentially affected in kidney disease [29].

## Materials and methods

### Participants

Data were obtained from the Alzheimer's Disease Neuroimaging Initiative (ADNI) database (http://adni.loni.usc.edu/). The ADNI was launched in 2003 as a public-private partnership, led by Principal Investigator Michael W. Weiner, MD. The primary goal of ADNI has been to test whether serial magnetic resonance imaging, positron emission tomography, other biological markers, and clinical and neuropsychological assessment can be combined to measure the progression of MCI and early Alzheimer's disease. The ADNI includes men and women with normal cognition, MCI and dementia between the ages of 55 and 90 from 63 sites in the United States and Canada. The ADNI excluded participants with significant systemic illness, unstable medical conditions, major depression and baseline brain imaging with focal lesions or multiple lacunae and thus comprised of a healthier cohort of older adults without underlying comorbidities other than cognitive impairment. The ADNI began in October 2004 and recruited participants in phases; ADNI-1 had 800 participants; 200 with Alzheimer's disease, 400 with MCI, and 200 with normal cognition. ADNI-GO had 200 participants with early amnestic MCI. ADNI 2 had another 650 participants. All ADNI participants from different phases of the ADNI study, (ADNI-1, ADNI-GO, and ADNI-2) with a baseline serum creatinine measurement were included in this analysis. Participants from ADNI-3 were excluded as they did not have a serum creatinine measurement.

We grouped participants with MCI [30] and dementia together as the group with cognitive impairment. Per the ADNI protocol, participants in this group with cognitive impairment had a) a subjective memory concern as reported by the participant, study partner, or clinician b) an abnormal memory function documented by scoring within the education adjusted ranges on the Logical Memory II Subscale, and c) Clinical Dementia Rating (CDR) $\geq$ 0.5 [31]. Participants in the control group had to be free of memory complaints (verified by a study partner) beyond what one would expect for age, have normal memory function documented by scoring above education adjusted cutoffs on the Logical Memory II subscale and a CDR score of 0 with a memory box score of 0.

### Measurement of kidney function (Independent variable)

Participating centers collected baseline serum creatinine measurement from all participants. We calculated the eGFR using the CKD Epidemiology Collaboration (CKD-EPI) equation [32] since CKD-EPI is more selective in classifying persons as having CKD and more accurately predicts the risk of vascular events, mortality and end-stage kidney disease [33, 34]. We categories participants by eGFR in the following groups; eGFR <45, 45–60, 61–90 and >90 ml/ min/ 1.73 m$^2$. For sensitivity analysis, we used the Modification of Diet in Renal Disease

Study equation (MDRD) [35] since MDRD is the most widely used equation for estimating eGFR in clinics and most clinical laboratories automatically report eGFR calculated by MDRD [36].

## Assessment of cognition (Outcome)

The ADNI uses a comprehensive battery of standard neuropsychological tests for the assessment of cognition. We used previously developed and validated composite scores for memory (ADNI-Mem) [37] and executive function (ADNI-EF) [38] for our analysis. ADNI-Mem is derived from Rey Auditory Verbal Learning Test (RAVLT, 2 versions), AD Assessment Schedule—Cognition (ADAS-Cog, 3 versions), Mini-Mental State Examination (MMSE), and Logical Memory data. ADNI-EF is derived from WAIS-R Digit Symbol Substitution, Digit Span Backwards, Trails A and B, Category Fluency, and Clock Drawing. We used the two composite scores as they incorporate all indicators of memory or executive function, maximizing measurement precision and account for version effects in some of the individual tests used in the ADNI. Moreover, these composite scores have linear scaling properties, tested validity and a good prediction of who would progress to MCI or dementia.

## Other variables

Baseline demographics; age, sex, race, ethnicity, marital status, years of education, body mass index (BMI), functional Activities Questionnaire (FAQ) score [39] and handedness were also obtained from the ADNI database.

## Statistical analysis

We evaluated baseline participant clinical and sociodemographic characteristics using descriptive statistics. We tested differences in baseline characteristics by eGFR categories (<45, 45–60, 61–90 and >90 ml/min/1.73 m$^2$) using linear trend tests for continuous variables and chi-square tests for categorical variables. We used the spearman correlations and scatterplots to describe the unadjusted association between eGFR (as a continuous variable) and ADNI-mem and ADNI-EF scores. After confirming normal distribution of ADNI-Mem and ADNI-EF scores, we ran multivariable linear regression models predicting cognition measured by ADNI-Mem and ADNI-EF scores as outcome variables and eGFR, age, sex, race, education and BMI as exposure variables (based on current literature on factors affecting cognition) to assess the adjusted association of eGFR (as a categorial variable and as a continuous variable) with cognition. We also performed a similar multivariable linear regression for participants with and without CKD defines as eGFR <60 ml/min/1.73 m$^2$. Since the majority of participants were either Caucasians or African Americans, we grouped race into Caucasian, African American and other races for the multivariable linear regression.

   To avoid confounding by baseline cognitive impairment status, we did sub-group analysis by grouping participants into groups with and without cognitive impairment. Baseline characteristics were compared using student's t-test and chi-square tests as appropriate. We analyzed the distribution of ADNI-mem and ADNI-EF scores in the two groups by eGFR categories described above. We repeated the multivariable linear regression analysis above in these sub-groups. We also performed a sub-group analysis for ages <75 and ≥75, men and women and Caucasians and non-Caucasians among participants with and without cognitive impairment. For sensitivity analysis with used eGFR calculated with the MDRD equation and repeated the multivariable linear regression analysis in the entire cohort and in groups with and without cognitive impairment [40, 41].

All analyses were done with SAS 9.4 (Cary, NC) with a p-value of 0.05 marking statistical significance.

## Results

The analysis included 1181 ADNI participants with baseline serum creatinine values available, 6% with eGFR<45, 22% with eGFR 45–60, 51% with eGFR 60–90 and 21% with eGFR>90 ml/min/1.73 m$^2$. Baseline characteristics are summarized in **Table 1**. There was no difference in age, ethnicity, years of education, BMI, FAQ score or handedness across the different categories of eGFR. More participants with lower eGFR were Caucasians and were widowed. More participants with a higher eGFR had cognitive impairment based on subjective complains, logical Memory II score and CDR score as described above. Lower eGFR groups did not have lower ADNI-Mem or ADMI-EF scores.

**Table 1. Baseline characteristics of included Alzheimer's Disease Neuroimaging Initiative (ADNI) participants in different ranges of estimated glomerular filtration rate.** Data are presented as mean and standard deviation for continuous variables and n (%) for categorical variables.

| | Total Sample (N = 1181) | eGFR (ml/min/1.73 m$^2$) | | | | p Value |
|---|---|---|---|---|---|---|
| | | <45 (n = 68) | 45–60 (n = 264) | 61–90 (n = 599) | >90 (n = 250) | |
| Age in years, mean ± SD | 73.7 ± 7.1 | 73.2 ± 7.3 | 73.7 ± 6.8 | 74.0 ± 7.0 | 73.4 ± 7.7 | 0.79 |
| Male, n (%) | 665 (56.3) | 16 (23.5) | 72 (27.3) | 367 (61.3) | 210 (84.0) | <0.001 |
| Ethnicity (%) | | | | | | 0.22 |
| Not Hispanic/Latino | 1146 (97.0) | 67 (98.5) | 252 (95.5) | 582 (97.2) | 245 (98.0) | |
| Hispanic/Latino | 29 (2.5) | 1 (1.5) | 8 (3.0) | 15 (2.5) | 5 (2.0) | |
| Unknown | 6 (0.5) | 0 (0.0) | 4 (1.5) | 2 (0.3) | 0 (0.0) | |
| Race n (%) | | | | | | 0.001 |
| Caucasian | 1097 (92.9) | 66 (97.1) | 252 (95.5) | 559 (93.3) | 220 (88.0) | |
| African American | 48 (4.1) | 1 (1.5) | 4 (1.5) | 19 (3.2) | 24 (9.6) | |
| American Indian/ Alaskan Native | 3 (0.3) | 0 (0.0) | 2 (0.8) | 1 (0.2) | 0 (0.0) | |
| Asian | 20 (1.7) | 0 (0.0) | 2 (0.8) | 13 (2.2) | 5(2.0) | |
| Native Hawaiian/ Other Pacific Islander | 2 (0.2) | 0 (0.0) | 1 (0.4) | 1 (0.2) | 0 (0.0) | |
| More than one race | 11 (0.9) | 1 (1.5) | 3 (1.1) | 6 (1.0) | 1 (0.4) | |
| Marital status n (%) | | | | | | 0.005 |
| Married | 896 (75.9) | 46 (67.6) | 182 (68.9) | 467 (78.0) | 201 (80.4) | |
| Widowed | 150 (12.7) | 16 (23.5) | 44 (16.7) | 67 (11.2) | 23 (9.2) | |
| Divorced | 98 (8.3) | 5 (7.4) | 26 (9.8) | 49 (8.2) | 18 (7.2) | |
| Never Married | 35 (3.0) | 1 (1.5) | 12 (4.5) | 16 (2.7) | 6 (2.4) | |
| Unknown | 2 (0.2) | 0 (0.0) | 0 (0.0) | 0 (0.0) | 2 (0.8) | |
| Years of education, mean ± SD | 15.9 ± 2.9 | 15.7 ± 2.9 | 15.8 ± 2.8 | 15.9 ± 2.9 | 16.1 ± 2.9 | 0.28 |
| BMI, mean ± SD | 26.6 ± 4.2 | 27.8 ± 6.2 | 26.5 ± 4.6 | 26.4 ± 3.9 | 26.9 ± 4.0 | 0.30 |
| FAQ score, mean ± SD | 4.7 ± 7.1 | 5.9 ± 8.6 | 5.1 ± 7.7 | 4.5 ± 7.0 | 4.6 ± 6.2 | 0.11 |
| Right-handed | 1082 (91.6) | 62 (91.2) | 245 (92.8) | 548 (91.5) | 227 (90.8) | 0.89 |
| Cognitive Impairment, n (%) | 805 (68.2) | 42 (61.8) | 169 (64.0) | 398 (66.4) | 196 (78.4) | 0.001 |
| ADNI-mem score, mean ± SD | 0.241 ± 0.874 | 0.390 ± 0.909 | 0.337 ± 0.935 | 0.250 ± 0.882 | 0.075 ± 0.750 | 0.004 |
| ADNI-EF score, mean ± SD | 0.160 ± 1.026 | 0.238 ± 1.040 | 0.213 ± 0.983 | 0.132 ± 1.053 | 0.151 ± 1.004 | 0.423 |
| Serum creatinine, mean ± SD | 1.00 ± 0.3 | 1.6 ± 0.3 | 1.2 ± 0.2 | 1.00 ± 0.1 | 0.7 ± 0.1 | <0.001 |
| eGFR (CKD-EPI), mean ± SD | 76.4 ± 19.7 | 40.1 ± 7.6 | 58.5 ± 8.8 | 77.8 ± 11.7 | 101.7 ± 7.6 | <0.001 |
| eGFR (MDRD), mean ± SD | 69.3 ± 17.4 | 35.9 ± 5.6 | 50.9 ± 5.0 | 71.9 ± 8.9 | 91.4 ± 6.8 | <0.001 |

BMI; body mass index (kg/m$^2$), FAQ; functional activities questionnaire, eGFR; estimated glomerular filtration rate, CKD EPI; Chronic Kidney Disease Epidemiology Collaboration, MDRD; The Modification of Diet in Renal Disease; ADNI-mem; composite memory score, ADNI-EF; composite executive function score.

Unadjusted correlation analysis showed a weak negative association between eGFR and ADNI-Mem scores and a statistically non-significant weak negative association between eGFR and ADNI-EF scores with correlation coefficients of -0.105 (p = 0.001) and -0.02, p = 0.43) respectively (**Fig 1**). Multivariable linear regression analysis (**Table 2**) showed an inverse relationship of age with ADNI-EF scores. Female sex was associated with higher ADNI-Mem scores and a higher education was associated with higher ADNI-Mem and ADNI-EF scores.

a)

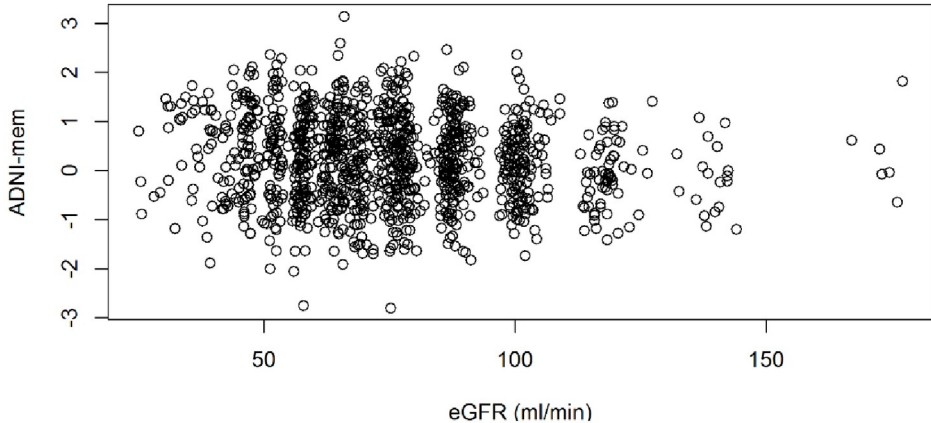

b)

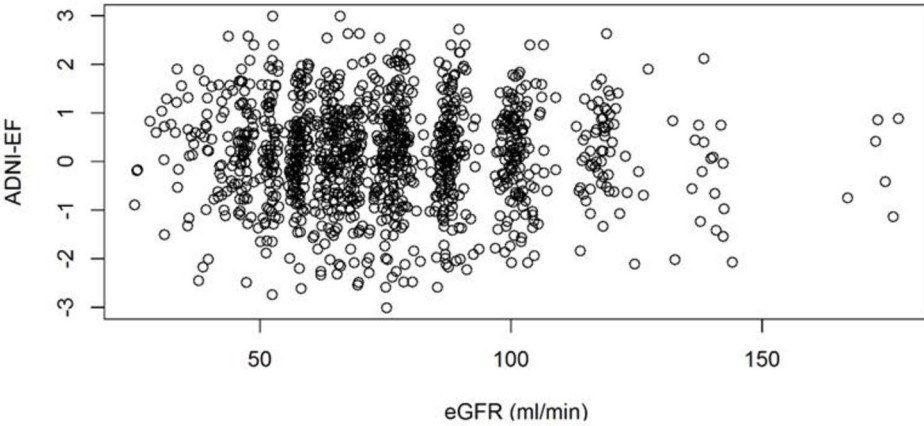

For ADNI-Mem correlation coefficient r = -0.105 and p=0.001; and for ADNI-EF r = -0.02, p= 0.43. ADNI-mem; composite memory score, ADNI-EF; composite executive function score, eGFR; estimated glomerular filtration rate using CKD-EPI equation

**Fig 1.** Scatterplot showing unadjusted correlation for a) ADNI-mem and b) ADNI-EF scores (y axis) and GFR (x-axis).

**Table 2. Multivariable linear regression model predicting ADNI-Mem score and ADNI-EF scores. A**. With eGFR as a categorical variable. Participants with eGFR <45, 45–60 and >90 are compared to participants with eGFR 61–90 ml/ min/1.73 m² taken as the reference group. **B**. With eGFR as a continuous variable.

| | Beta estimate for ADNI-Mem score | 95% confidence interval | p value | Beta estimate for ADNI-EF score | 95% confidence interval | p value |
|---|---|---|---|---|---|---|
| **A** | | | | | | |
| Age (+10) | -0.07 | -0.16, 0.03 | 0.17 | -0.24 | -0.35, -0.13 | <0.0001 |
| Female sex | 0.36 | 0.20, 0.51 | <0.0001 | 0.17 | -0.01, 0.35 | 0.27 |
| AA race | 0.25 | -0.10, 0.60 | 0.16 | -0.20 | -0.60, -0.20 | 0.32 |
| Other race | -0.06 | -0.50, 0.37 | 0.77 | 0.02 | -0.49, 0.52 | 0.95 |
| Years of education (+1) | 0.08 | 0.06, 0.10 | <0.0001 | 0.11 | 0.08, 0.13 | <0.0001 |
| BMI (+1) | 0.01 | 0, 0.03 | 0.08 | 0.01 | -0.01, 0.03 | 0.23 |
| GFR <45 | -0.09 | -0.42, 0.25 | 0.61 | 0.03 | -0.36, 0.42 | 0.88 |
| GFR 45–60 | 0.08 | -0.10, 0.26 | 0.40 | 0.10 | -0.11, 0.31 | 0.33 |
| GFR >90 | -0.06 | -0.24, 0.12 | 0.51 | 0.11 | -0.1, 0.31 | 0.32 |
| **B** | | | | | | |
| Age (+10) | -0.06 | -0.16,0.03 | 0.19 | -0.25 | -0.36, -0.14 | <0.0001 |
| Female sex | 0.37 | 0.22, 0.53 | <0.0001 | 0.20 | 0.02, 0.37 | 0.03 |
| AA race | 0.24 | -0.11, 0.58 | 0.18 | -0.19 | -0.59, 0.21 | 0.35 |
| Other race | -0.07 | -0.5, 0.37 | 0.77 | 0.02 | -0.48, 0.52 | 0.95 |
| Years of education (+1) | 0.08 | 0.06, 0.10 | <0.0001 | 0.11 | 0.08, 0.13 | <0.0001 |
| BMI (+1) | 0.01 | 0,0.03 | 0.09 | 0.01 | -0.01, 0.03 | 0.21 |
| eGFR (+10) | -0.01 | -0.04, 0.02 | 0.56 | 0.01 | -0.03, 0.05 | 0.69 |

ADNI-Mem; ADNI composite memory score, ADNI-EF; ADNI composite executive function score, AA; African American, BMI; body mass index (kg/m²), eGFR; estimated glomerular filtration rate (ml/min/1.73 m²). For race, African American race and other races were compared with Caucasian race.

There was no association between eGFR and ADNI-Mem or ADNI-EF scores. **S1 Table of S1 File** shows the same analysis comparing participants with and without CKD (eGFR <60 ml/ min/1.73 m² and eGFR ≥60 ml/ min/1.73 m²) with similar results.

When categorized by cognitive impairment (with and without cognitive impairment) **(S2 Table of S1 File)**, the mean eGFR was higher in the group with cognitive impairment. The distribution of eGFR was similar in both groups **(S1 Fig of S1 File)**. ADNI-Mem and ADNI-EF scores were similar across the eGFR categories in both groups **(S3 Table of S1 File)**. No significant correlation was seen between eGFR and ADNI-Mem or ADNI-EF scores in participants with cognitive impairment. In participants without cognitive impairment, a weak negative correlation was seen between eGFR and ADNI-mem scores while no significant correlation was seen between eGFR and ADNI-EF scores **(S2 Fig of S1 File)**. Separate multivariable linear regression analysis in the groups with and without cognitive impairment **(S4 Table of S1 File)** showed similar associations for age, sex, and education and eGFR with ADNI-Mem and ADNI-EF scores. There was no association between eGFR and ADNI-Mem or ADNI-EF scores, even after stratification by age, sex and race **(S5 Table of S1 File)**. Our results remained consistent when we used eGFR calculated with MDRD instead of CKD-EPI equation **(S6 Table of S1 File)**.

## Discussion

We aimed to assess the association between eGFR and cognition in ADNI participants. We showed that modestly low eGFR was not associated with lower memory or executive function

in the older adults from the ADNI cohort. In adjusted analysis, male sex was associated with lower memory scores while older age was associated with lower executive function scores. Lower education was associated with both lower memory and executive function scores. Lower eGFR was not associated with either memory or executive function.

Using the ADNI cohort had the advantage of a detailed evaluation of cognition with a comprehensive battery of standard neuropsychological tests and validated composite scores for memory and executive function, the two main domains of cognition typically affected in kidney disease [29]. We used validated composite scores for memory and executive function rather than results from individual neuropsychological tests to conserve the statistical power by reducing the number of potential comparisons and to reduce measurement error. In addition, these composite scores were built with consideration for the variation in results that may be present due to use of different versions of some neuropsychological tests used in the ADNI [37].

Moreover, the ADNI had relatively healthy older adults. Lower eGFR in older adults may have different implications than in younger adults. Among persons with eGFR levels <45 ml/min/1.73 $m^2$ at baseline, older adults are less likely (than their younger counterparts) to experience an annual decline in eGFR of >3 ml/min/1.73 $m^2$ [14] or progression to ESKD [15]. The development of ESKD is a much rarer event in older compared with younger patients with an eGFR of 30 to 59 mL/min/1.73 $m^2$ [16]. Similarly, biopsy studies indicate that unlike progressive glomerulosclerosis in the young, age associated glomerulosclerosis does not contribute to progressive CKD [17]. Although there is a high prevalence of cognitive impairment in CKD [42, 43], this association may not be applicable to older adults [44]. This difference may be secondary to age related decline in eGFR in the absence of an actual kidney 'disease' pathology or inaccurate categorization of older adults as CKD due to imperfect estimation of GFR by commonly used serum creatinine-based equations.

Our results are consistent with the results from the BRain IN Kidney disease (BRINK) study where eGFR >30 mL/min per 1.73 $m^2$ was not associated with cognitive impairment [19]. Although smaller, this study like the ADNI, used a comprehensive battery of neuropsychological tests to assess memory and executive function. The Health ABC study [28] did not show an association between eGFR and cognition (odds ratio 1.10, 95% CI, 0.80 to 1.51) in participants >73 years supporting the results of our study. Similarly, the French 3C study [20], the Norwegian HUNT study [21] or the Australian Sydney Memory and Ageing Study [22] did not show an detrimental association between lower eGFR and cognitive decline or dementia.

Some other studies have shown a graded decrease in cognition with lower GFR. Although these studies are meritorious in their own right, there are some important differences between these studies and our study that may account for the contrasting results. Some studies included patients with severe kidney disease with lower eGFR and/or younger participants [23–25, 44] while others had inadequate matching at baseline and/or limited neuropsychological testing with use of tests meant for screening to assess global cognition. The single center study by Kurella et al. [24] compared patients with CKD and ESKD to published normative values matched for age and education. The study population was younger (mean age 62±14.3 years) and had more severe kidney disease (mean serum creatinine 3.1 ± 1.9 mg/dl and mean eGFR 18.7 ± 35.3 mL/min per 1.73 $m^2$) than our cohort. The REGARDS study [23] used a 6- item cognitive screening test incorporated into the baseline telephone interview. Although easier to perform in a large study such as REGARDS with over 20,000 participants, screening tests especially when over phone may lack the specificity of comprehensive neuropsychological tests. In addition, screening tests assess global cognition and not specific domains of cognition such as memory and executive function that are preferentially affected in CKD. Phone based cognitive

tests can be affected by hearing loss in older adults. Moreover, baseline differences in age and vascular disease with CKD patients being older and with greater burden of comorbidities than controls. Since age and comorbid conditions are risk factors for cognitive impairment, these differences can lead to positive confounding. Similarly, the CRIC-COG study [25] also had differences in baseline characteristics where participants with lower eGFR had lower education and greater burden of comorbidities. Another study from the Third National Health and Nutrition Examination Survey (NHAHES) [45] had younger participants with a mean age of 36; the results may thus not be extrapolatable to older adults.

Our study has limitations. We performed a cross-sectional analysis, and therefore causality or directionality cannot be inferred. The ADNI data did not have serial measurements of kidney function. Only 6% of the participants had an eGFR of <45 mL/min/1.73m$^2$. This was expected as severe CKD was an exclusion criterion for the study. However, since we wanted to explore the association between eGFR and cognition in older adults without known progressive intrinsic kidney disease, this cohort was well suited for our study as age associated decline in eGFR is not expected to cause severe CKD with eGFR <45 mL/min/1.73m$^2$. We used serum creatinine based eGFR measurements. Although serum creatinine-based calculation of eGFR is standard clinical practice, changes in muscle mass rather than kidney function can change eGFR and eGFR may not correlate well with measured GFR in persons with normal kidney function. Also, sarcopenia is associated with both cognitive impairment and CKD and cause confounding. This may explain the higher eGFR in the group with cognitive impairment in our study. Although we did not have results from the gold standard dual-energy X-ray absorptiometry for measurement of muscle mass, we did compare BMIs across the eGFR groups and did not find any differences.

In conclusion, we showed that mild-moderate CKD in older adults (likely attributable to physiological aging) is not associated with cognitive impairment. With almost half of the population over age 70 years with CKD [7], our findings have important clinical implications and provide clinically useful information for physicians that can guide management and education of older patients at highest risk of dementia.

## Supporting information

**S1 File.**
(DOCX)

**S2 File.**
(PDF)

## Acknowledgments

¶Data used in preparation of this manuscript were obtained from the Alzheimer's Disease Neuroimaging Initiative (ADNI) database (adni.loni.usc.edu). The ADNI investigators contributed to the design and implementation of the ADNI and/or provided data but did not participate in analysis or writing of this report. A complete listing of ADNI investigators can be found at: http://adni.loni.usc.edu/wp-content/uploads/how_to_apply/ADNI_Acknowledgement_List.pdf

**Presented** at the American Society of Nephrology (ASN) week 2019.

## Author Contributions

**Conceptualization:** Aditi Gupta.

**Formal analysis:** Kevin Kennedy.

**Methodology:** Aditi Gupta, Jaime Perales-Puchalt, Srinivasan Beddhu, Mark Sarnak, Jeffrey Burns.

**Writing – original draft:** Aditi Gupta.

**Writing – review & editing:** Aditi Gupta, Jaime Perales-Puchalt, David Drew, Srinivasan Beddhu, Mark Sarnak, Jeffrey Burns.

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
