## [Decision Letter · Decision Letter 0]

13 Aug 2020

PONE-D-20-20489

Mild to moderate CKD is not associated with cognitive impairment in the elderly

PLOS ONE

Dear Dr. Gupta,

Thank you for submitting your manuscript to PLOS ONE. After careful consideration, we feel that it has merit but does not fully meet PLOS ONE’s publication criteria as it currently stands. Therefore, we invite you to submit a revised version of the manuscript that addresses the points raised during the review process.

Two experts raised important issues.  Normal vs declined eGFR should be analyzed.  Also you claimed this article is focus on aged cohort but the cohort selection is not clearly described in regards of age.

We look forward to receiving your revised manuscript.

Kind regards,

Tatsuo Shimosawa, M.D., Ph.D.

Academic Editor

PLOS ONE

Journal Requirements:

2. Please amend either the title on the online submission form (via Edit Submission) or the title in the manuscript so that they are identical.

3. One of the noted authors is a group or consortium Alzheimer’s Disease Neuroimaging Initiative. In addition to naming the author group, please list the individual authors and affiliations within this group in the acknowledgments section of your manuscript. Please also indicate clearly a lead author for this group along with a contact email address.

4. Thank you for including the following funding information within the acknowledgements section of your manuscript; " Data collection and sharing for this project was funded by the Alzheimer's Disease Neuroimaging Initiative (ADNI) (National Institutes of Health Grant U01 AG024904) and DOD ADNI (Department of Defense award number W81XWH-12-2-0012). ADNI is funded by the National Institute on Aging, the National Institute of Biomedical Imaging and Bioengineering, and through generous contributions from the following: AbbVie, Alzheimer’s Association; Alzheimer’s Drug Discovery Foundation; Araclon Biotech; BioClinica, Inc.; Biogen; Bristol-Myers Squibb Company; CereSpir, Inc.; Cogstate; Eisai Inc.; Elan Pharmaceuticals, Inc.; Eli Lilly and Company; EuroImmun; F. Hoffmann-La Roche Ltd and its affiliated company Genentech, Inc.; Fujirebio; GE Healthcare; IXICO Ltd.; Janssen Alzheimer Immunotherapy Research & Development, LLC.; Johnson & Johnson Pharmaceutical Research & Development LLC.; Lumosity; Lundbeck; Merck & Co., Inc.; Meso Scale Diagnostics, LLC.; NeuroRx Research; Neurotrack Technologies; Novartis Pharmaceuticals Corporation; Pfizer Inc.; Piramal Imaging; Servier; Takeda Pharmaceutical Company; and Transition Therapeutics. The Canadian Institutes of Health Research is providing funds to support ADNI clinical sites in Canada. Private sector contributions are facilitated by the Foundation for the National Institutes of Health (www.fnih.org). The grantee organization is the Northern California Institute for Research and Education, and the study is coordinated by the Alzheimer’s Therapeutic Research Institute at the University of Southern California. ADNI data are disseminated by the Laboratory for Neuro Imaging at the University of Southern California. "

"This work is supported by NIH K23 AG055666 (AG) Cognitive Impairment in End Stage Renal Disease, NIH CTSA grant UL1 TR000001 (KUMC), and NIH P30 AG035982 (KU ADC)."

Additionally, because some of your funding information pertains to [commercial funding//patents], we ask you to provide an updated Competing Interests statement, declaring all sources of commercial funding.

In your Competing Interests statement, please confirm that your commercial funding does not alter your adherence to PLOS ONE Editorial policies and criteria by including the following statement: "This does not alter our adherence to PLOS ONE policies on sharing data and materials.” as detailed online in our guide for authors  http://journals.plos.org/plosone/s/competing-interests.  If this statement is not true and your adherence to PLOS policies on sharing data and materials is altered, please explain how.

Please include the updated Competing Interests Statement and Funding Statement in your cover letter. We will change the online submission form on your behalf.

Reviewers' comments:

Reviewer's Responses to Questions

**Comments to the Author**

1. Is the manuscript technically sound, and do the data support the conclusions?

Reviewer #1: Partly

Reviewer #2: Yes

2. Has the statistical analysis been performed appropriately and rigorously? 

Reviewer #1: No

Reviewer #2: Yes

3. Have the authors made all data underlying the findings in their manuscript fully available?

Reviewer #1: No

Reviewer #2: Yes

4. Is the manuscript presented in an intelligible fashion and written in standard English?

Reviewer #1: Yes

Reviewer #2: Yes

5. Review Comments to the Author

Reviewer #1: To examine whether the relationship between CKD and dementia hold true in the elderly, the authors conducted a cross-sectional secondary analysis of an established observational cohort of data from the Alzheimer’s Disease Neuroimaging Initiative (ADNI) including a non-probability sample of all ADNI participants with serum creatinine measurements at baseline. The authors showed that modestly low eGFR was not associated with lower memory or executive function in the elderly population of the ADNI cohort using multivariable linear regression analysis.

The theme of this study is intriguing and the manuscript is well-written; however, there are some concerns to be addressed.

1. It is considered that eGFR correlates with the measured GFR in patients with renal dysfunction, while it is doubtful that the same correlation is shown in patients with normal renal function. How would you describe that point?

2. BMI should be entered into all multivariable models.

3. I wonder whether the decorrelation between eGFR and cognition in the multivariable linear regression analysis would be maintained when stratified by eGFR (eGFR<60 or eGFR ≥60). The authors should show the results stratified by eGFR.

4. The authors describe “Unadjusted correlation analysis showed weak negative associations of ADNI-Mem scores and ADNI-EF scores with eGFR with correlation coefficients of -0.105 (p=0.001) and -0.02, p=0.43) respectively (Figure 1)” in Page 8 and “Weak correlation between eGFR and ADNI-Mem and ADNI-EF scores were again seen (Supplementary Figure 2)” in Page 10 in the Results section. These descriptions are incorrect because some of these associations are not significant. The authors should revise these descriptions.

Reviewer #2: This is a well done study examining the association of eGFR with cognitive impairment from the Alzheimers Disease Neuroimaging initiative. I have only a few minor suggestions for improvement.

Could the authors include more details on entry criteria for the parent study, I don't see age criteria mentioned, perhaps include age range to clarify whether the cohort was restricted to older adults or not? Much of the rationale for why the authors chose to conduct this study in this particular cohort does not come out until the discussion. I believe the argument is that this is an older cohort without much underlying comorbidity other than dementia or mild cognitive impairment for some? Presenting this in the introduction could provide a stronger rationale for why the authors chose to conduct this study in this particular cohort.

In the introduction, it would be good to present more detail on the demographic characteristics of study populations in which the association between eGFR and cognitive function has been measured. The introduction makes it sound like there have been no prior studies of cognitive impairment and eGFR in older populations? My sense is that some prior studies have been conducted in older adults, so summarizing those studies and their results would be helpful then highlighting original contribution of this study (it sounds like the contribution of this study is that it included mostly healthy older adults?)

Consider less sweeping title, the authors found no association of eGFR in this particular cohort, but I am not sure this justifies the statement that there is no association in "the elderly." Maybe no association in older adults enrolled in the Alzheimers Disease Neuroimaging initiative?

6. PLOS authors have the option to publish the peer review history of their article (what does this mean?). If published, this will include your full peer review and any attached files.

Reviewer #1: No

Reviewer #2: **Yes: **ann ohare

---

## [Author Response · Author response to Decision Letter 0]

27 Aug 2020

We thank the editors and the reviewers for their thoughtful feedback. We are also thankful for their suggestions which we believe have improved the manuscript. Our response to each concern is outlined below.

Editors: 

1) Normal vs declined eGFR should be analyzed. 

Response: We have added another multivariable linear regression analysis with this stratification of eGFR (Supplementary Table 1).

2) You claimed this article is focus on aged cohort but the cohort selection is not clearly described in regards of age.

Response: We agree. We have clarified the age under methods and described the cohort in more detail under introduction in the revised manuscript.

Reviewer #1

1) It is considered that eGFR correlates with the measured GFR in patients with renal dysfunction, while it is doubtful that the same correlation is shown in patients with normal renal function. How would you describe that point?

Response: We agree and have clarified this further in our limitations. Obtaining measured GFR is cumbersome and impractical, eGFR is often used in both research and clinical practice. Since our study is essentially a null study and there is no association between any level of eGFR and cognition (Figure 1, Table 2), this does not alter our results or conclusion. 

2) BMI should be entered into all multivariable models.

Response: We have added BMI in the multivariate models. 

3) I wonder whether the decorrelation between eGFR and cognition in the multivariable linear regression analysis would be maintained when stratified by eGFR (eGFR<60 or eGFR ≥60). The authors should show the results stratified by eGFR.

Response: We have added this stratification of eGFR (Supplementary Table 1).

4) The authors describe “Unadjusted correlation analysis showed weak negative associations of ADNI-Mem scores and ADNI-EF scores with eGFR with correlation coefficients of -0.105 (p=0.001) and -0.02, p=0.43) respectively (Figure 1)” in Page 8 and “Weak correlation between eGFR and ADNI-Mem and ADNI-EF scores were again seen (Supplementary Figure 2)” in Page 10 in the Results section. These descriptions are incorrect because some of these associations are not significant. The authors should revise these descriptions.

Response: Thank you for pointing this out. We have revised the descriptions as suggested. 

Reviewer #2: 

1) Could the authors include more details on entry criteria for the parent study, I don't see age criteria mentioned, perhaps include age range to clarify whether the cohort was restricted to older adults or not? 

Response: We have added the mean age (74) in ADNI in the introduction and the age range (55-90) in the methods section where we describe the ADNI cohort. For ADNI 2 the minimum age for the normal control group and the significant memory concern group was 65 years. 

2) Much of the rationale for why the authors chose to conduct this study in this particular cohort does not come out until the discussion. I believe the argument is that this is an older cohort without much underlying comorbidity other than dementia or mild cognitive impairment for some? Presenting this in the introduction could provide a stronger rationale for why the authors chose to conduct this study in this particular cohort.

Response: We agree. We have added this detail in the last paragraph of the introduction in the revised manuscript. 

3) In the introduction, it would be good to present more detail on the demographic characteristics of study populations in which the association between eGFR and cognitive function has been measured. The introduction makes it sound like there have been no prior studies of cognitive impairment and eGFR in older populations? My sense is that some prior studies have been conducted in older adults, so summarizing those studies and their results would be helpful then highlighting original contribution of this study (it sounds like the contribution of this study is that it included mostly healthy older adults?)

Response: We agree and have added more detail about other published studies as suggested.

4) Consider less sweeping title, the authors found no association of eGFR in this particular cohort, but I am not sure this justifies the statement that there is no association in "the elderly." Maybe no association in older adults enrolled in the Alzheimer’s Disease Neuroimaging initiative? 

Response: Thank you for this suggestion. We have changed the title to ‘Mild-moderate CKD is not associated with cognitive impairment in older adults in the Alzheimer’s Disease Neuroimaging Initiative cohort’ to more accurately describe this study.

---

## [Decision Letter · Decision Letter 1]

15 Sep 2020

Mild-moderate CKD is not associated with cognitive impairment in older adults in the Alzheimer’s Disease Neuroimaging Initiative cohort

PONE-D-20-20489R1

Dear Dr. Gupta,

We’re pleased to inform you that your manuscript has been judged scientifically suitable for publication and will be formally accepted for publication once it meets all outstanding technical requirements.

Kind regards,

Tatsuo Shimosawa, M.D., Ph.D.

Academic Editor

PLOS ONE

Additional Editor Comments (optional):

Reviewers' comments:

Reviewer's Responses to Questions

**Comments to the Author**

1. If the authors have adequately addressed your comments raised in a previous round of review and you feel that this manuscript is now acceptable for publication, you may indicate that here to bypass the “Comments to the Author” section, enter your conflict of interest statement in the “Confidential to Editor” section, and submit your "Accept" recommendation.

Reviewer #1: All comments have been addressed

Reviewer #2: All comments have been addressed

2. Is the manuscript technically sound, and do the data support the conclusions?

Reviewer #1: Yes

Reviewer #2: Yes

3. Has the statistical analysis been performed appropriately and rigorously? 

Reviewer #1: Yes

Reviewer #2: Yes

4. Have the authors made all data underlying the findings in their manuscript fully available?

Reviewer #1: Yes

Reviewer #2: Yes

5. Is the manuscript presented in an intelligible fashion and written in standard English?

Reviewer #1: Yes

Reviewer #2: Yes

6. Review Comments to the Author

Reviewer #1: The authors' responses to my comments were appropriate and to the point. I have no further comments.

Reviewer #2: The authors have addressed my comments, this should be a nice contribution to the literature. I do not have additional comments on the current draft.

7. PLOS authors have the option to publish the peer review history of their article (what does this mean?). If published, this will include your full peer review and any attached files.

Reviewer #1: No

Reviewer #2: **Yes: **ann o'hare

---

## [Editor Report · Acceptance letter]

29 Sep 2020

PONE-D-20-20489R1 

Mild-moderate CKD is not associated with cognitive impairment in older adults in the Alzheimer’s Disease Neuroimaging Initiative cohort

Dear Dr. Gupta:

I'm pleased to inform you that your manuscript has been deemed suitable for publication in PLOS ONE. Congratulations! Your manuscript is now with our production department. 

Kind regards, 

on behalf of

Prof. Tatsuo Shimosawa 

Academic Editor

PLOS ONE